# The Antimicrobial, Antioxidant, and Anticancer Activity of Greenly Synthesized Selenium and Zinc Composite Nanoparticles Using *Ephedra aphylla* Extract

**DOI:** 10.3390/biom11030470

**Published:** 2021-03-22

**Authors:** Mustafa Mohsen El-Zayat, Mostafa M. Eraqi, Hani Alrefai, Ayman Y. El-Khateeb, Marwan A. Ibrahim, Hashim M. Aljohani, Maher M. Aljohani, Moustafa Mohammed Elshaer

**Affiliations:** 1Unit of Genetic Engineering and Biotechnology, Faculty of Science, Mansoura University, Mansoura City 35516, Egypt; mzayat2001@yahoo.com; 2National Research Center, Department of Microbiology and Immunology, Veterinary Research Division, Dokki Giza 12622, Egypt; m.eraqi@mu.edu.sa; 3Department of Biology, College of Science, Majmaah University, Majmaah 11952, Saudi Arabia; 4Medical Biochemistry Department, Faculty of Medicine, Mansoura University, Mansoura City 35516, Egypt; 5Department of Internal Medicine, Infectious Diseases Division, College of Medicine, University of Cincinnati, Cincinnati, OH 45267, USA; 6Department of Agricultural Chemistry, Faculty of Agriculture, Mansoura University, Mansoura City 35516, Egypt; aymanco@mans.edu.eg; 7Department of Zoology, Women’s College, Ain Shams University, Cairo City 11566, Egypt; 8Department of Molecular Genetics and Biochemistry, College of Medicine, University of Cincinnati, Cincinnati, OH 45221, USA; aljohahm@mail.uc.edu; 9Department of Clinical Laboratory Sciences, College of Applied Medical Sciences, Taibah University, Medina City 42353, Saudi Arabia; 10Department of Pathology, College of Medicine, Taibah University, Medina City 42353, Saudi Arabia; Maher.aljohani@yahoo.com; 11Department of Pathology and Laboratory Medicine, Ministry of The National Guard-Heath Affairs, Medina City 42353, Saudi Arabia; 12Department of Microbiology at Specialized Medical Hospital, Mansoura University, Mansoura City 35516, Egypt; moustafaelshaer@mans.edu.eg

**Keywords:** *Ephedra aphylla*, aqueous extract, selenium, zinc, nanoparticles, antioxidant, oxidation, cytotoxic, antimicrobial, cancer

## Abstract

The current work aimed to synthesize selenium and zinc nanoparticles using the aqueous extract of *Ephedra aphylla* as a valuable medicinal plant. The prepared nanoparticles were characterized by TEM, zeta potential, and changes in the phytochemical constituents. Hence, the phenolic, flavonoid, and tannin contents were reduced in the case of the prepared samples of nanoparticles than the original values in the aqueous extract. The prepared extract of *Ephedra aphylla* and its selenium and zinc nanoparticles showed high potency as antioxidant agents as a result of the DPPH^•^ assay. The samples were assessed as anticancer agents against six tumor cells and a normal lung fibroblast (WI-38) cell line. The selenium nanoparticles of *Ephedra aphylla* extract revealed very strong cytotoxicity against HePG-2 cells (inhibitory concentration (IC_50_) = 7.56 ± 0.6 µg/mL), HCT-116 cells (IC_50_ = 10.02 ± 0.9 µg/mL), and HeLa cells (IC_50_ = 9.23 ± 0.8 µg/mL). The samples were evaluated as antimicrobial agents against bacterial and fungal strains. Thus, selenium nanoparticles showed potent activities against Gram-negative strains (*Salmonella typhimurium*, *Pseudomonas aeruginosa*, *Klebsiella pneumoniae*, and *Escherichia coli*), Gram-positive strains *(Bacillus cereus, Listeria monocytogenes, Staphylococcus aureus*, and *Staphylococcus epidermidis*), and the fungal strain *Candida albicans*. In conclusion, the preparation of nanoparticles of either selenium or zinc is crucial for improved biological characteristics.

## 1. Introduction

Plants are rich in different types of natural compounds. About 25% of the prescription products in the world originated from wild or cultivated plants [1]. Recent research in synthetic chemistry reported a good potential of natural compounds to provide better aspects of treatment and prevention of many diseases [2,3,4,5]. Plant-derived anticancer drugs, such as vincristine, vinblastine, camptothecin, and taxol, are a part of the battle against tumor cells [1]. The continuing search for new antitumor natural compounds is a promising avenue for its prevention or treatment [6]. Plant compounds such as alkaloids, phenolics, flavonoids, phenyl-propanoids, and terpenoids have also been reported to have anticancer activity [7,8].

Green synthesis of nanomaterials using plant extracts as a furious source of natural compounds like carbohydrates, phenolics, flavonoids, tannins, and alkaloids [9,10] that could act as safe reducing and stabilizing agents in addition to their ability to maintaining aseptic environments throughout the process [11,12]. Therefore, medicinal plants of therapeutic potential could control the dimensions and forms of the biosynthesized nanomaterial [13,14]. Selenium and zinc are vital trace elements in living organisms that play an important role in antioxidant defense, immune regulation, and antitumor for human health. Biosynthesized zinc and selenium nanoparticles play a great advantage due to their higher degradability, lower toxicity, and ability to clear from the body [15,16,17,18].

*Ephedra* is the only genus in the family Ephedraceae. It comprises 40 species distributed worldwide. In Egypt, the genus *Ephedra* is represented by five species [19]. They are characterized by their medicinal importance. Many of these were applied in traditional medicine for treating bronchial disorders and asthma [20,21].

*E. aphylla* is distributed along the eastern Mediterranean region up to the Arabian Peninsula. It is a large shrub that grows in the cracks of limestone cliffs or nearby valleys in the sand and usually grows in juniper forests with *Pistacia*, *Opuntia*, *Daphne linearifolia*, *Artemesia,* and *Thymelaea hirsuta*. *E. aphylla* is present in the Egyptian protected area of Wadi El-Gemal-Hamata in the National Reserve of Egypt [22].

*Ephedra aphylla* has been reported as being rich with valuable phytochemicals such as alkaloids like ephedrine, pseudoephedrine, N-methylephedrine, and 6-methoxykynurenic acid, and hordenine [22,23], phenolics and flavonoids, i.e., di-*C*-glucosyl flavone, 2,2′-di-*O*-β-glucopyranosyl-vicenin, di-*O*-glycoside, herbacetin 3-*O*-α -rhamnopyranoside-8-*O*-β-glucopyranoside, vicenin, 7-methoxy-4-oxo-1,4-dihydroquinoline-2-carboxylic acid, ephedralone, 4-hydroxybenzoic acid, (*E*)-3-(4-hydroxyphenyl)acrylic acid, 3,4-dihydroxybenzoic acid, and 7-methoxy-herbacetin [24,25].

Some of the phytochemical constituents in *Ephedra aphylla* possess larvicidal activity and could be used for controlling disease-caused by mosquitoes [26]. *E. aphylla* showed strong antiproliferative potential against the breast cancer cell lines MFC7 and T47D that may be assigned to the active phytochemicals in the plant, such as ephedrine alkaloids and herbacetin [27]. Ephedrine has been described to suppress hepatocyte growth factor (HGF)-induced cancer cell motility by preventing both HGF-induced phosphorylation of c-Met and its tyrosine kinase activity [28]. It was also reported that *Ephedra* extracts exhibit antimetastatic and antitumor effects by suppressing the hepatocyte growth factor-c-Met signaling pathway through the inhibition of c-Met tyrosine kinase activity [28].

The intent of the current work was to biosynthesize selenium and zinc nanoparticles of the aqueous extract of *Ephedra aphylla* as a source of reducing components, and to assess their biological characteristics as antioxidant, anticancer, and antimicrobial agents.

## 2. Materials and Methods

### 2.1. Materials

#### 2.1.1. Chemicals and Reagents

Folin-Ciocalteau reagent (Fluka, Biochemical Inc., Bucharest, Romania), Gallic acid (Biomedical Inc., Orange City, FL, USA), 1,1-Diphenyl-2-picrylhydrazyl (DPPH^•^), aluminum chloride, sodium hydroxide, sodium nitrite, catechin, vanillin, hydrochloric acid, ascorbic acid, (3-(4,5-dimethylthiazol-2-yl)-2,5-diphenyltetrazolium bromide) (MTT), RPMI-1640 medium, and DMSO were purchased from Sigma Aldrich (St. Louis, MO, USA). Sodium Carbonate (El-Nasr Pharmaceutical Chemicals, Cairo, Egypt). Fetal Bovine serum (FBS) (GIBCO, UK). ZnSO4 (Andenex-Chemie, Hamburg, Germany), SeSO4 (Alpha Chemika, Panvel, Maharashta, India). Doxorubicin, Ceftazidime and Ampicillin-Sulbactam were purchased from Merck (Darmstadt, Germany).

#### 2.1.2. Plant Materials

The stems of *Ephedra aphylla* were collected from their authentic habitats at Saint Catherine Protectorate, South Sinai, Egypt. The plant was taxonomically identified by Mustafa El-Zayat and authenticated by Boulos [29].

### 2.2. Phytochemical Analysis

The active constituents were extracted using the same trend of usage in folklore medicine as hot infusions. Twenty grams of the dried plant stems were mixed with 200 mL deionized water with shaking for 30 min in a water bath system at 70 °C. The produced extract was filtered, and the filtrate was kept at 4 °C for further use.

The total phenolic, flavonoid, and tannin components of the aqueous extract of *Ephedra aphylla* stems were quantitatively estimated.

#### 2.2.1. Total Phenolic Contents

The total phenolic constitutes were assessed by the Folin-Ciocalteu procedure progressed by Wolfe et al. [30,31], in which the procedure involved the use of gallic acid as a standard. The investigated values of total phenolic constitute in the aqueous extract of *Ephedra aphylla*, and its nanoparticles with zinc and selenium were quantified as equivalents in milligram of gallic acid/dried plant extract in gram using a standard curve (y = 0.0062x, r^2^ = 0.987).

#### 2.2.2. Total Flavonoid Contents

The total flavonoids constitutes were appreciated by a colorimetric assessment using aluminum chloride as conveyed by Zhishen et al. [32] with the use of catechin as a standard. The values of total flavonoid constitutes were quantified as equivalents of catechin in milligram per dried plant extract in gram using a standard curve (y = 0.0028x, r^2^ = 0.988).

#### 2.2.3. Total Tannin Contents

The total tannins constitutes were assessed using vanillin-hydrochloride assay [33,34], and the values of the anticipated samples were quantified as equivalents of tannic acid in gram/dried plant in 100 g.

### 2.3. Preparation of Metal Nanoparticles

The procedure from Devasenan et al. [35] was applied with an insignificant amendment in the synthesis of the metal nanoparticles. Each selenium sulfate (1 mmol) or zinc sulfate (1 mmol) was dissolved in deionized water (20 mL), and the solution was added gradually in portions to a well-stirred plant extract (20 mL). After the complete addition of the metal aqueous solution, the mixture was stirred for an extra 2 h at room temperature. The formed nanoparticles were acquired in the equimolar ratio in both cases.

### 2.4. Structure Characterization of the Metal Nanoparticles

#### 2.4.1. Transmission Electron Microscope (TEM)

The physical properties and chemical structure, i.e., particle’s size, shape, surface nature, crystal structure, and morphological data of the prepared nanoparticles, were identified as conveyed by Otunola et al. [36] using TEM (JEOL TEM-2100, Tokyo, Japan) at the Electron Microscope Unit, Mansoura University, Egypt. The analysis was run with a 200 nm magnification value.

#### 2.4.2. Nanoparticles Characteristic via Zeta Potential

The surface charge of the prepared selenium and zinc nanoparticles in the suspension was characterized by applying Zeta potential technique using Malvern Instruments Ltd. Zeta Potential Ver. 2.3 (Kassel, Germany) according to Bhattacharjee, [37] at the Electron Microscope Unit, Mansoura University, Egypt. The process is significant for studying the surface nature of nanoparticles, and the stability of these particles can be expected to last for long-term periods [38].

### 2.5. Potential Biological Applications

#### 2.5.1. Antioxidant Activity—DPPH Assay

The antioxidant capacity of the aqueous extract of *Ephedra aphylla* and its selenium and zinc nanoparticles was investigated following the DPPH^•^ colorimetric method using ascorbic acid as a standard by way of the assay reported by Kitts et al. [39]. The serial dilution of each sample was prepared by mixing the sample with methanol in an equivalent amount. The DPPH^•^ solution was prepared in a concentration of 0.135 mM and mixed with each sample in the serial dilution with a volume of 1 mL. after the addition of DPPH^•^ solution; the samples were kept in the dark for 30 min at room temperature. The absorbance of each sample was measured at 517 nm in the next step. The % DPPH^•^ remaining was calculated using Equation (1):% DPPH^•^ remaining = [DPPH^•^]_T_/[DPPH^•^]_T = 0_ × 100(1)

The values of % DPPH^•^ remaining were plotted versus mg extract/mL using an exponential curve to identify the inhibitory concentration “IC_50_”. IC_50_ indicates the constitutes amount of antioxidants needed to decrease the initial concentration of DPPH^•^ solution by 50%. The values of IC_50_ point out the inverse relationship with the antioxidant capacity of the tested sample [40].

#### 2.5.2. Antimicrobial Activity Procedure

The antimicrobial potential of the studied *Ephedra aphylla* extract and the prepared zinc and selenium nanocomposites were estimated using the agar well diffusion assay [41,42]. Ceftazidime (CAZ) and Ampicillin-Sulbactam (SAM) were used as standard antibiotics. The tested microbial and fungal species were *Salmonella typhimurium* (ATCC^®^ 14028™), *Pseudomonas aeruginosa* (ATCC^®^ 9027™), *Staphylococcus epidermidis* (ATCC^®^ 12228™), *Klebsiella pneumonia* (ATCC^®^ 10031™), *Bacillus cereus* (ATCC^®^ 11778™), *Staphylococcus aureus* (ATCC^®^ 6538™), *Escherichia coli* (ATCC^®^ 10536™), *Listeria monocytogenes* (ATCC^®^ 19115™), and *Candida albicans* EMCC number-105. The obtained strains were of animal origin and obtained from the Microbiological Resources Centre (MIRCEN), Faculty of Agriculture, Ain Shams University.

#### 2.5.3. Anticancer Activity Procedure

Six human tumor cell lines specifically; Hepatocellular carcinoma (HePG-2), Epithelioid cervix carcinoma (Hela), Epidermoid larynx carcinoma (HEP2), Human prostate cancer (PC3), Mammary gland carcinoma (MCF-7), and Colorectal carcinoma (HCT-116), were acquired from the ATCC holding company for biological products and vaccines (VACSERA), Cairo, Egypt. Doxorubicin was used as a stock chemotherapeutic anticancer drug. The chemical reagents RPMI-1640 medium, MTT, DMSO (Sigma co., St. Louis, MO, USA), Fetal Bovine Serum (FBS; Gibco Life Technologies, Paisley, UK).

Standard colorimetric MTT assay was applied to assess the cytotoxicity of the investigated samples by measuring the cell growth following the procedure conveyed by Bondock et al. [43]. Concisely, the reduction in the yellow color of MTT (2-(4,5-dimethylthiazol-2-yl)-3,5-diphenyl-2*H*-tetrazol-3-ium bromide) to purple formazan was achieved by mitochondrial succinate dehydrogenases of living cells. The cell strains were grown in RPMI-1640 medium with 10% fetal bovine serum. Antibiotics, penicillin (100 units/mL) and streptomycin (100 µg/mL) were added at 37 °C in a 5% CO_2_ incubator. Cell lines were seeded in a 96-well plate at a density of 1.0 × 10^4^ cells/well at 37 °C for 48 h under 5% CO_2_. After incubation, cells were treated with different concentrations of the tested samples and incubated for an additional 24 h. After 24 h of drug treatment, MTT solution (5 mg/mL, 20 µL) was then added and incubated over again for 4 h. DMSO (100 µL) was then added to each well to dissolve the produced violet formazan. The colorimetric evaluation was reverent, and the absorbance values were measured at 570 nm utilizing a plate reader (EXL 800, New York, NY, USA). The IC_50_ values were calculated using nonlinear regression (sigmoid type), analyzed using the Origin 8.0^®^ software (OriginLab Corporation, https://www.originlab.com/ (accessed on 19 March 2021)). Relative cell viability in percent was calculated from Equation (2):% Relative cell viability = [Sample absorbance/Control absorbance] × 100(2)

## 3. Results and Discussion

The chemical constitutes of aqueous stem extract of *Ephedra aphylla*, and its sources comprise a wide range of diverse privileged secondary metabolites, in which they are potentially reducing materials for the biogenic production of nanoparticles [44]. The present study demonstrated that *Ephedra aphylla* stem extract is filled with phenolics (131.55 mg gallic acid equivalent/g dry extract), flavonoids (27.51 mg catechin equivalent/g dry extract), and tannins (64.91 mg gallic acid equivalent/g dry extract) that could be utilized for the reduction and stabilization of selenium and zinc metal ions and the green synthesis of their nanoparticles. Polyphenol compounds have an electron resonance hybrid effect, as they play the role of biological reduction in salts ions and convert them into nanoparticles, as well as having a role in stabilizing those particles in a stable, non-precipitating form [45,46]. Flavonoids as a subclass of phenolics are difficult to break, and therefore they are used in the bio-reduction in zinc and selenium ions, and their conversion into nanoparticles where the flavonoids aggregate and bind on the surface of the nanoparticles and neutralize their charges to zero-valent molecules in the nanometer range, and thus new compounds are formed that have very small sizes, thus increasing their surface area and are active, effective, and unique, chemically and biologically [45,46,47]. Regarding the prepared nano selenium and nano zinc, there were marked decreases in the phenolics (26.85 and 40.63 mg gallic acid equivalent/g dry extract, respectively), flavonoids (7.09 and 2.98 mg catechin equivalent/g dry extract, respectively), and tannins (15.82 and 15.05 mg gallic acid equivalent/g dry extract, respectively) (Table 1).

### 3.1. Characterization of the Prepared Nanoparticles

#### 3.1.1. Transmission Electron Microscope (TEM)

TEM technique was applied for characterizing digital images of the prepared selenium and zinc nanoparticles of *Ephedra aphylla*, which provided a good view of morphological particles. The samples were analyzed at a higher spatial resolution (200 nm). Figure 1 shows the TEM micrographs and size distributions of the prepared selenium and zinc nanoparticles. Figure 1a shows the formation of the spherical and tetragonal shapes of the selenium nanoparticles, in addition to zinc nanoparticles (Figure 1b), which seemed to be spherical particles only, which indicated the crystallinity of the particles, supported by the emission diffraction of the selected area. The size of the selenium particles is ranged from 13.95 to 26.26 nm, while the size of the zinc particles is ranged from 8.34 to 15.20 nm. TEM allowed the assessment of agglomeration and/or aggregation of the constituent nanoparticles. The zinc nanoparticles were more aggregated than the selenium nanoparticle providing a large surface area of selenium nanoparticles that would increase their efficiency when applied as a cytotoxic agent.

#### 3.1.2. Zeta Potential Analysis

The significance of this analysis lies in the possibility of studying the nature of the particles present on the surface of the nanoparticles, and thus, these particles could be expected to be stable long-term. It was found that the nanoparticles had a charge on their surface, being able to attract a thin layer of ions opposite them in the charge, and from here, this analysis was applied with the zeta potential technique to define the nature of the charge on the surface of the nanoparticles. The nanoparticles contained a double layer of ions which were transferred as they diffused into solution. The electrical potentials at the borders of the double layer were defined as the zeta potential of the particles and had values ranging from +100 mV to −100 mV. The selenium and zinc particles synthesized with the *Ephedra aphylla* extract had zeta potential values of −5.61 and −8.78 mV (Figure 2), which were highly stable because nanoparticles with zeta potential values less than +25 mV or greater than −25 mV generally have a high degree of stabiliity as described by Honary and Zahir [38].

### 3.2. Biological Potentials

#### 3.2.1. Antioxidant Activity

The water extract of the *Ephedra aphylla* stems showed an IC_50_ of 0.053 mg/mL, while the zinc and selenium nano-solution of *Ephedra aphylla* extract showed lower antioxidant activity with IC_50_ values of 0.213 and 0.296 mg extract/mL that was compared to ascorbic acid as a strong antioxidant with an IC_50_ of 0.022 mg extract/mL.

The obtained results indicated that *Ephedra aphylla* extract was rich in phenolics, flavonoids, and tannins that possess reasonable antioxidant activity and are well known for their antioxidant potential based mainly on their structure, especially the number and attitude of the hydroxyl groups that are of importance for green synthesis as reducing agents due to their ability to reduce ions into nanoparticles [48,49]. Utilizing the groups responsible for antioxidant activity in the extract during biosynthesis led to a decrease in antioxidant activity.

#### 3.2.2. Anticancer Activity

The anticancer activities of polar and nonpolar extracts of naturally occurring *E. aphylla* against MFC7 and T47D cell lines have been recently reported [28]. It was reported that nano-selenium and nano-zinc in their zero-oxidation state exhibit lower toxicity and excellent bioavailability and could be stabilized by encapsulation into suitable nano-vehicles [50,51]. They have a wide range of medical applications like antioxidant properties, the capability of reducing oxidative stress [52], chemopreventive activity as a potential anticancer drug, and antimicrobial effects [53,54]. The hollow spherical selenium nanoparticles (SeNPs) reduce the risk of selenium toxicity [55]. The results of many studies indicate that Nano-Se can be more helpful in cancer chemoprevention as a potential anticancer drug [56,57] as well as an anticancer drug delivery carrier [58]. Moreover, the immunostimulatory effect of nanoscale selenium has been confirmed [59]. SeNPs have also shown remarkable anticancer activity [60,61,62] and exhibit high potential in cancer chemotherapy and as drug carriers [63]. The anticancer effects of SeNPs are mediated through their ability to inhibit the growth of cancer cells through the induction of cell cycle arrest at the S phase [56]. Cell membrane plays an important role in SeNPs-induced toxicity in cancer cells. SeNPs treatment changes the biomechanical properties of cancer cells; in particular, they remarkably decrease the adhesion force and Young’s modulus [64]. Besides unique anticancer efficacy, SeNPs have been proved to present a better selectivity between normal and cancer cells [65].

Besides their direct anticancer effects, SeNPs have been pointed to as potential anticancer drug delivery carriers [56]. A key factor that usually contributes to nanomaterial-based drug cytotoxicity is cellular uptake [56]. The nano-size of these materials allows an efficient uptake by various cell types and selective drug accumulation at target sites [66]. Zinc nanoparticles have great potential in cancer therapy. It has been reported that ZnNPs induce selective killing of cancer cells where ZnNPs selectively induce apoptosis in cancer cells, which is likely to be mediated by reactive oxygen species via the p53 pathway, through which most anticancer drugs trigger apoptosis [67].

In this work, the cytotoxic activities of the prepared *Ephedra aphylla* extract and its selenium and zinc nanoparticles were evaluated using an MTT assay. The samples were tested in vitro against six tumor cells, i.e., HepG-2, MCF-7, HCT-116, PC3, HeP2, and HeLa cell lines. Doxorubicin was selected as a reference drug, comparing the results of the tested samples against the different cancer cells. IC_50_ values express the concentrations that induced 50% of the death of tumor cells in µg/mL. The IC_50_ values are inversely proportional to the efficiency of the sample to inhibit the growth of the cancer cells. Therefore, a potent cytotoxic agent would require the lowest concentration and IC_50_ values. 

The results of in vitro cytotoxicity are listed in Table 2, in which the scale of the cell’s viability or potency of the samples was mentioned in Table 2. The results demonstrated, in general, that the prepared selenium and zinc nanoparticles had potent cytotoxicity against the diverse tumor cell lines than the original extract. Accordingly, the nanoparticles had a large surface size that increased the efficiency of the sample to inhibit the growth of the tumor cells. Moreover, the selenium nanoparticles of *Ephedra aphylla* revealed the most potent cytotoxicity against HePG-2 cell lines with an IC_50_ of 7.56 ± 0.6 µg/mL. Besides, strong cytotoxicity was recorded for selenium nanoparticles of *Ephedra aphylla* with an IC_50_ of 15.65 ± 1.4 µg/mL against MCF-7 tumor cell lines, indicating high effectiveness of the selenium nanoparticles in the first order, accompanied by those of zinc nanoparticles that had moderate cytotoxicity against MCF-7 cell lines with an IC_50_ of 29.32 ± 2.2 µg/mL. Selenium nanoparticles of *Ephedra aphylla* revealed strong cytotoxicity against HCT-116 cancer cell lines with an IC_50_ of 10.02 ± 0.9 µg/mL. On the other hand, the tested samples were less potent anticancer agents against PC3 cancer cell lines, but selenium nanoparticles of *Ephedra aphylla* remained strong cytotoxic agents with an IC_50_ of 18.63 ± 1.5 µg/mL. Selenium nanoparticles of *Ephedra aphylla* also showed strong cytotoxicity against HeP2 and HeLa tumor cell lines with an IC_50_ of 12.10 ± 1.2, and 9.23 ± 0.8 µg/mL, respectively. In addition, all the tested samples revealed weak cytotoxic activities against the normal lung fibroblast (WI-38) cell line. The results indicated the potential for the application of these samples as anticancer drugs. Briefly, the *Ephedra aphylla* seemed to be an effective cytotoxic agent against all the tested tumor cell lines as used as a simple extract or its isolated zinc or selenium nanoparticles. The results showed that using selenium metal ions to prepare nanoparticles of the extract of *Ephedra aphylla* had higher efficiency in inhibiting the growth of cancer cells more than the use of zinc metal ions.

The results of selenium and zinc sulfate solutions (Table 2) demonstrated that both samples displayed moderate activities against HePG-2 cell lines (IC_50_ = 32.98 ± 2.3 and 36.75 ± 1.9), HeP2 cell lines (IC_50_ = 39.04 ± 1.9 and 44.1 ± 2.1), and HeLa cell lines (IC_50_ = 33.26 ± 1.6 and 34.62 ± 1.8) relative to the results of the reference standard, extracted sample, and the prepared nanoparticles. Besides, the solutions of selenium and zinc sulfate revealed relatively weak cytotoxicity against MCF-7, HCT-116, and PC3 tumor cell lines. The results of the investigated salt solutions reflect the efficiency of the extracted *Ephedra aphylla*, and its selenium and zinc nanoparticles on the cytotoxicity in comparison with the results of the salt solution samples. Concisely, the salts, in general, did not impact the cytotoxicity against the tested tumor cell lines.

A comparative relationship of the results of the extract of *Ephedra aphylla* and its selenium and zinc nanoparticles against HepG-2, MCF-7, HCT-116, PC3, HeP2, and HeLa cell lines with the results of Doxorubicin is specified in Figure 3. It is worth mentioning that the formation of metal nanoparticles enhanced the cytotoxic characters of the extract of *Ephedra aphylla* by different ranges. The difference in the results between selenium and zinc nanoparticles or the extract of *Ephedra aphylla* itself depended on the type of tumor cell lines or the nature of the nanoparticles of the used metal ions, as it was characterized as having small particles and higher aggregation. Generally, the extract of *Ephedra aphylla* was an efficient cytotoxic agent against HeLa cell lines with effective cytotoxicity (IC_50_ = 23.92 ± 1.7 µg/mL) when compared to the other tested cell lines. The selenium nanoparticle of the extract of *Ephedra aphylla* was the most potent cytotoxic agent against the other tested samples and its results against the diverse cell lines (IC_50_ = 9.23 ± 0.8 µg/mL). The potent cytotoxicity of zinc nanoparticles was also noted against HePG-2 cell lines (IC_50_ = 17.46 ± 1.1 µg/mL). Subsequently, the extract of *Ephedra aphylla* and its metal nanoparticles were applicable for the inhibition of cancer cell growth against HePG-2 and HeLa cell lines rather than the other tested tumor cells.

Figure 4 shows the inhibition % of the *Ephedra aphylla* extract and its metals nanoparticles against all the tested human tumor cells at different concentrations. The samples were prepared as a serial dilution of seven concentrations starting from 100 µg/mL. The results demonstrated that the use of a higher concentration of a sample increased the efficiency of the sample to inhibit the growth of the cancer cells. In consequence, the extract of *Ephedra aphylla* displayed potent cytotoxicity with inhibition percentages ranging from 63.6% to 76.4% at a concentration of 100 µg/mL, while the use of 1.56 µg/mL of the same *Ephedra aphylla* extract resulted in non-cytotoxic characteristics against all the tested cancer cell lines. The prepared selenium nanoparticles of *Ephedra aphylla* verified the strongest potency against all the tested cancer cell lines with inhibition percentages ranging from 76.3% to 91.7% at the higher concentration, but it remains has weak activities at the lower concentration (1.56 µg/mL) against HePG-2, HCT-116, HeP2, and Hela tumor cell lines. The results of zinc nanoparticles of the extracted *Ephedra aphylla* demonstrated the capability of the nanoparticles to improve the anticancer behavior of the extract against all the tested cancer cell lines. The percent of inhibition was also calculated for the selenium and zinc sulfate solutions at different concentrations (1.56–100 µg/mL). All the solutions of the metal salts displayed moderate activities at the higher level of concentrations (100 µg/mL) with the percentage of inhibition ranging from 40.4% to 48.5% for the selenium sulfate solution, and 33.7% to 40.4% for the zinc sulfate solution against diverse tumor cell lines. The results of the percentage of inhibition agreed with that of the plant extract and its selenium and zinc nanoparticles.

Figure 4 shows the percent of average relative viability of all tumor cell lines at different concentrations. The percent viability of tumor cells is inversely proportional to the inhibition percentage at the same concentration. The results were matched with that recorded for the *Ephedra aphylla* extract and its selenium and zinc nanoparticles against diverse human tumor cell lines. The results of average relative viability of tumor cell lines showed non-cytotoxic effects at lower concentrations 1.56–12.5 µg/mL for both solutions of metal salts. The average relative viability of both solutions of metal salts at the higher concentration (100 µg/mL) demonstrated weak cytotoxic effects of both solutions of metal salts against all human tumor cell lines with a cell viability average range from 51.5–59.6% for the solution of selenium sulfate, and 59.6–66.3% for the solution of zinc sulfate. The principal aspect affecting the results of cytotoxicity was the efficiency of the extracted *Ephedra aphylla*, and its selenium and zinc nanoparticles. It seemed obvious that the salt solutions had an inoperative impact on the results obtained.

#### 3.2.3. Antimicrobial Activity

The antimicrobial potential of *Ephedra aphylla* aqueous extract in addition to the prepared selenium and zinc nanoparticles solutions were tested against several pathogenic microbial isolates using an agar well diffusion assay, as presented in Table 3 and Figure 5. The *Ephedra aphylla* aqueous extract expressed no antimicrobial activity against any of the tested microbes. It was noted that selenium nanoparticles of the aqueous extract of *Ephedra aphylla* are potent antimicrobial agents, and their derived components, such as selenium sulfide, are mostly applied in medicine for the treatment of infectious diseases, e.g., *Malassezia* and *Tinea versicolor*. Nevertheless, the excess amount of selenium led to toxic effects and selenosis. Therefore, the current research was focused on reducing cell toxicity and develop the bio-functional characteristics of selenium. Consecutively, nanotechnology has provided a safe approach to decrease the toxicity and enhance the functionality of selenium over biosynthesis [68].

The prepared selenium nanoparticles displayed a broad and proficient antimicrobial impact against Gram-positive bacterial strains, i.e., *S. aureus, S. epidermis, B. cereus*, *L. monocytogenes,* Gram-negative bacterial strains, i.e., *P. aeruginosa, E. coli, S. typhimuriu*, and *K. pneumonia*, and fungal species, i.e., *C. albicans*, in which the results were compared with the standard antibiotics (Gentamycin, and Clotrimazole).

The higher concentration of nanoparticles of zinc displayed accomplished growth inhibition as a result of their impact on the growth of pathogenic bacterial species. Accordingly, it was proposed that at low concentrations of zinc, it behaved as bacteriostatic; however, at the higher concentrations, the material had bactericidal influence. The proposed discussion is in agreement with the results of numerous earlier researches [69,70].

As conveyed in the earlier reports, it was found that the nanoparticles of zinc demonstrated a wide range of antibacterial characteristics. Therefore, the antibacterial behavior was improved due to the reduced size of the particles and hence increased the surface area of the particles and increased its efficiency in inhibiting bacterial growth [71]. The mechanism of action, in this case, is still not clarified, and the antibacterial activity of zinc nanoparticles is a result of these points: (i) The cell membrane has electrostatic interaction with the nanoparticles of zinc; (ii) cellular internalization of zinc nanoparticles; (iii) The role and impact of reactive oxygen species such as phenolic components. Concisely, the interaction could occur between the zinc nanoparticles and the cell membrane with the change in the cell membrane permeability against the tested nanoparticles. Therefore, induced oxidative stress took place owing to the entrance of the nanoparticles into the cell, causing growth inhibition and cell death [72].

The prepared zinc nanoparticles exhibited a broad and efficient antimicrobial spectrum against Gram-positive bacteria, i.e., *S. aureus, B. cereus*, *L. monocytogenes*; Gram-negative *P. aeruginosa, E. coli, S. typhimurium*, *K. pneumonia*, and expressed no activity against the fungal species *C. albicans* and the Gram-positive bacteria *S. epidermidis*.

Nowadays, bacterial infections are a serious threat because of the antibiotic resistance crisis that has been correlated to the overuse and misuse of antibiotics in addition to the lack of new medications development. The development of antibiotic-resistant bacteria such as *S. aureus* has minimized the effectiveness of antibiotics for treatment and has increased the need for the discovery of new antimicrobial drugs of natural origin [73,74,75]. This study provided evidence for the synergetic relation between *Ephedra aphylla* extracts and the produced selenium and zinc nanoparticles, and the obtained results revealed that the resulted products from our study could be used as an alternative for antibiotics as it was clear that these products were more efficient than the compared antibiotics, in addition to their broad spectrum of antimicrobial activity against antibiotic-resistant bacteria such as *S. aureus*.

## 4. Conclusions

The present work provides a green technique for the synthesis of selenium and zinc nanoparticles from an extract of *Ephedra aphylla* stems. The selenium and zinc nanoparticles had higher stability and various desired morphologies due to the presence of certain chemical constituents such as phenolic, flavonoid, tannin, and alkaloids content that were responsible for nanoparticle biosynthesis and stability. The results of this study indicated that the prepared nano-solutions expressed potent antimicrobial and anticancer activities along with reduced antioxidant characters. The most potent cytotoxic results were recorded for the nano-selenium solution of the plant extract with high potency against HePG-2, MCF-7, HCT-116, PC3, HeP2, and Hela cell lines (IC50 = (7.56 ± 0.6)–(18.63 ± 1.5) µg/mL), relative to the reference standard along with high inhibition percentages and low relative viability at different concentrations (1.56–100 µg/mL). The selenium nanoparticles displayed potent antimicrobial activities against diverse bacterial and fungal species with improved inhibition zone diameter within 19.33–39.33 mm. The antioxidant characters and phytochemical constitutes were found to be more reduced than in the original extract. Hence, the nano-selenium solution of the extracted *Ephedra aphylla* stems could be widely applied in the future in various biological implementations in medicine, cosmetic manufacturing, food processing, and many other applications.

## Figures and Tables

**Figure 1 biomolecules-11-00470-f001:**
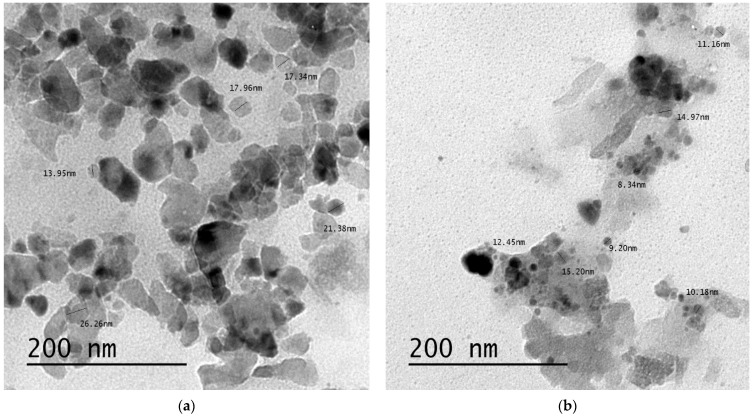
TEM micrographs of the prepared selenium and zinc nanoparticles of the extract of *Ephedra aphylla*. (**a**) TEM micrographs and size distributions for selenium nanoparticles synthesized by *Ephedra aphylla* extract at a 200 nm magnification value. (**b**) TEM micrographs and size distributions for zinc nanoparticles synthesized by *Ephedra aphylla* extract at a 200 nm magnification value.

**Figure 2 biomolecules-11-00470-f002:**
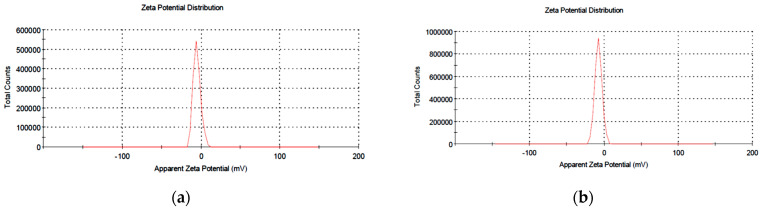
Zeta potential charts of the prepared selenium and zinc nanoparticles of the extract of *Ephedra aphylla*. (**a**) Zeta potential of the prepared nano selenium synthesized by *Ephedra aphylla* extract. (**b**) Zeta potential of the prepared nano zinc synthesized by *Ephedra aphylla* extract.

**Figure 3 biomolecules-11-00470-f003:**
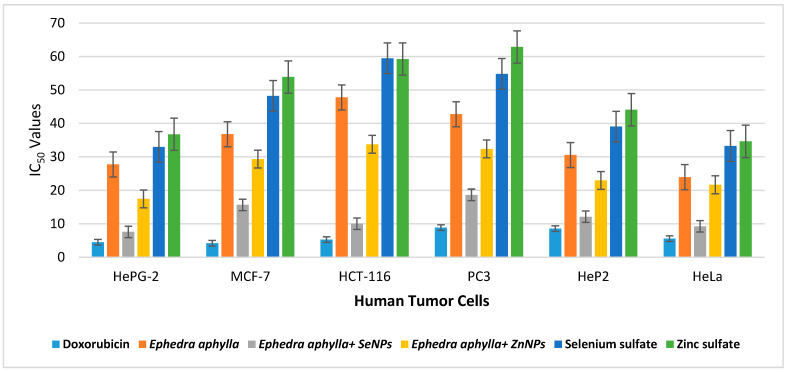
Comparison of the IC_50_ values of the tested samples against human cancer cells.

**Figure 4 biomolecules-11-00470-f004:**
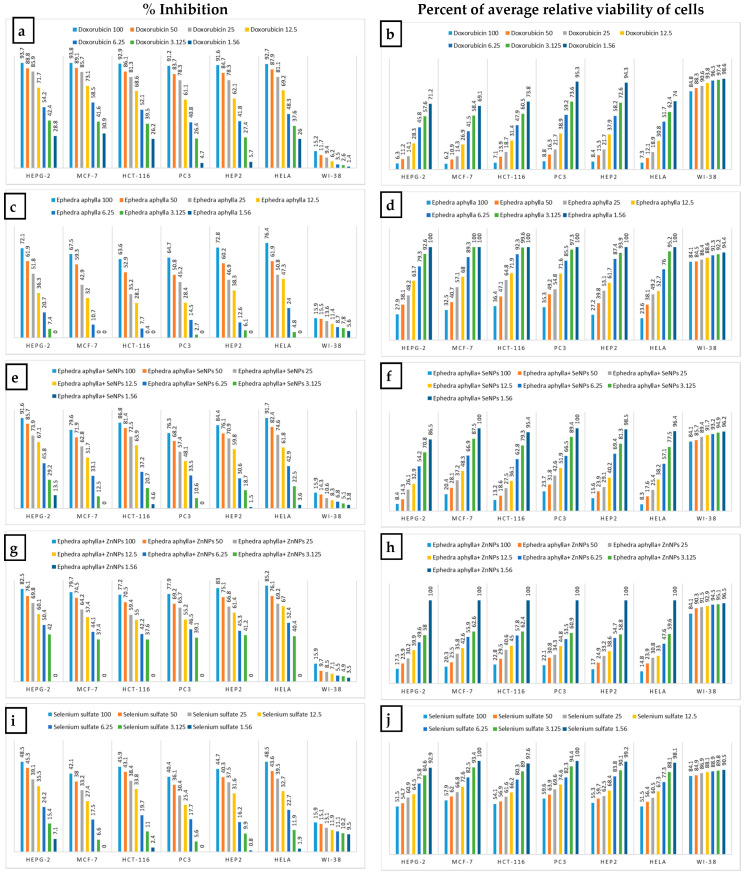
Comparison of the inhibition percentage with the percent of average relative viability of tumor and normal cells at different concentrations. Where: (**a**,**b**) for Doxorubicin, (**c**,**d**) for *Ephedra aphylla* extract, (**e**,**f**) for *Ephedra aphylla* + SeNPs, (**g**,**h**) for *Ephedra aphylla* + ZnNPs, (**i**,**j**) for Selenium sulfate solution and (**k**,**l**) for Zinc sulfate solution.

**Figure 5 biomolecules-11-00470-f005:**
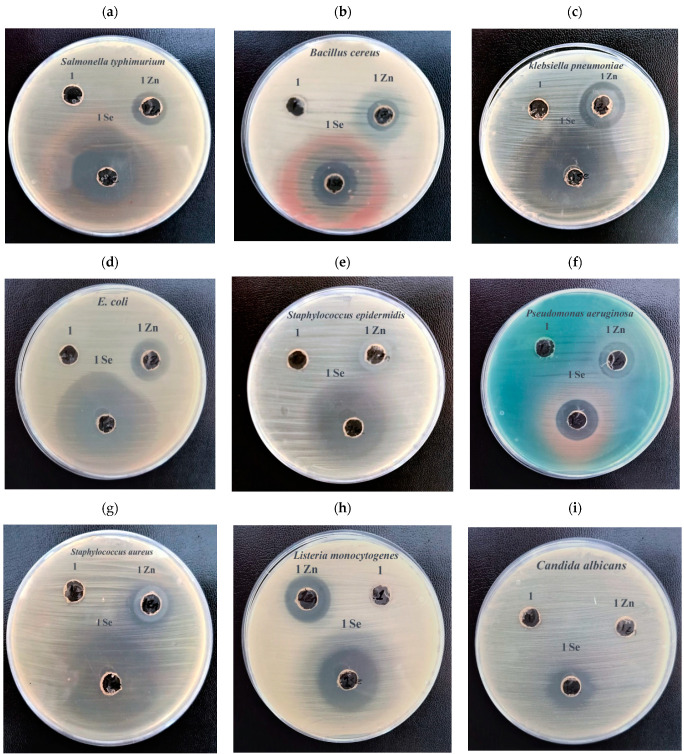
Photos of the antimicrobial activity of the wild *Ephedra aphylla* water extract and the synthesized nanoparticles using well diffusion assay against different pathogenic microbial starins as presented in subfigures (**a**): *Salmonella typhimurium*, (**b**): *Bacillus cereus*, (**c**): *Klebsiella pneumoniae*, (**d**): *Escherichia coli*, (**e**): *Staphylococcus epidermidis*, (**f**): *Pseudomonas aeruginosa*, (**g**): *Staphylococcus aureus*, (**h**): *Listeria monocytogenes* and (**i**): *Candida albicans* where, Code 1 = *Ephedra aphylla* aqueous extract; Code 1 Se = greenly synthesized selenium nanoparticles; Code 1 Zn = greenly synthesized zinc nanoparticles.

**Table 1 biomolecules-11-00470-t001:** The phytochemical analysis of *Ephedra aphylla* extract and its selenium and zinc nanoparticles (NPs).

Samples	Phytochemical Analysis		
	Phenolic Contents “mg Gallic Acid Equivalent/g Dry Extract”	Flavonoid Contents “mg Catechin Equivalent/g Dry Extract”	Tannin Contents “mg Gallic Acid Equivalent/g Dry Extract”
*Ephedra aphylla* extract	131.55	27.51	64.91
*Ephedra aphylla* + SeNPs	26.85	7.09	15.82
*Ephedra aphylla* + ZnNPs	40.63	2.98	15.05

**Table 2 biomolecules-11-00470-t002:** Cytotoxic activity of the prepared samples against the diverse human tumor cells.

Samples	In Vitro Cytotoxicity, IC_50_ ± SD (µg/mL) ^(a)^						
	HePG-2	MCF-7	HCT-116	PC3	HeP2	HeLa	WI-38
Doxorubicin	4.50 ± 0.2	4.17 ± 0.2	5.23 ± 0.3	8.87 ± 0.6	8.54 ± 0.6	5.57 ± 0.4	94.94
*Ephedra aphylla*	27.72 ± 2.1	36.77 ± 2.8	47.77 ± 3.3	42.76 ± 3.1	30.53 ± 2.4	23.92 ± 1.7	>100
*Ephedra aphylla*+ SeNPs	7.56 ± 0.6	15.65 ± 1.4	10.02 ± 0.9	18.63 ± 1.5	12.10 ± 1.2	9.23 ± 0.8	>100
*Ephedra aphylla*+ ZnNPs	17.46 ± 1.1	29.32 ± 2.2	33.74 ± 2.7	32.36 ± 1.9	22.95 ± 1.1	21.65 ± 1.8	96.76
Selenium sulfate	32.98 ± 2.3	48.24 ± 2.7	59.50 ± 3.1	54.83 ± 2.8	39.04 ± 1.9	33.26 ± 1.6	>100
Zinc sulfate	36.75 ± 1.9	53.89 ± 2.4	59.26 ± 3.3	62.86 ± 3.2	44.1 ± 2.1	34.62 ± 1.8	>100

^(a)^ IC_50_: inhibitory concentration (µg): 1–10 (very strong), 11–20 (strong), 21–50 (moderate), 51–100 (weak), and above 100 (non-cytotoxic).

**Table 3 biomolecules-11-00470-t003:** Antimicrobial activity of the greenly synthesized selenium nanoparticles using *Ephedra aphylla* stem extract on various pathogenic microbial strains.

Pathogenic Bacterial Strains	Inhibition Zones Measured in Millimeters ^(a)^				Standard Antibiotic
	Plant Extract	Nano-Zinc Composite	Nano-Selenium Composite	SAM	CAZ
**Gram-negative bacteria**					
*Salmonella typhimurium*	-	16	39.3	15	20
*Pseudomonas aeruginosa*	-	17	20	R	19
*Klebsiella pneumoniae*	-	21	38.3	15	28
*Escherichia coli*	-	20	47	15	24
**Gram-positive bacteria**					
*Staphylococcus epidermidis*	-	-	31	15	8
*Bacillus cereus*	-	14	21	13	7
*Staphylococcus aureus*	-	19	36.3	12	20
*Listeria monocytogenes*	-	20	26.7	34	13
**Fungi**					
*Candida albicans*	-	-	19.33	17	18

^(a)^ The diameter of the well (8.0 mm) is included in the measured zone of inhibition, Ceftazidime (CAZ) and Ampicillin-Sulbactam (SAM).

## Data Availability

Data is contained within the article and Appendix A.

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
