# Peer review of "The Antimicrobial, Antioxidant, and Anticancer Activity of Greenly Synthesized Selenium and Zinc Composite Nanoparticles Using Ephedra aphylla Extract"

_biomolecules, 2021, doi:10.3390/biom11030470_

Round 1

Reviewer 1 Report

I appreciate your eagerness to respond to the suggestions. I believe your paper and research will be of interest for many.

Author Response

Dear Reviewer, 1,

I am very glad to receive your letter, thanks for your time, effort, and support.

Yours.

Hani Alrefai

The corresponding author

Reviewer 2 Report

In the presented manuscript authors describe the antimicrobial, antioxidant, and anticancer activity of 2 greenly synthesized selenium and zinc composite nanoparticles 3 using Ephedra aphylla extract. The work is correct and well presented. The authors conducted numerous analytical and biological experiments of the obtained compounds. Despite this, some drawbacks can be noted:

  1. In cell line studies, no studies with normal cell lines were performed. The essence of substances with anti-cancer potential is relative selectivity in reference to normal cells.
  2. If the authors present a table (Table 2) with IC50 values and additionally present the same results on a graph, there is no need to add another table with raw data from which the IC50 was calculated (Table 3). This results in an artificial increase in the volume of the manuscript. Alternatively, Table 3 should be included in Supplementary data.
  3. In the line 457 the present simple tens should be used (... provide ...). 

Author Response

Dear Reviewer, 2,

I am very glad to receive your letter, thanks for your time, effort and support.

Please find below our point-by-point response for the manuscript biomolecules-1105418. We have successfully replied to all of your precious comments and revised the manuscript accordingly. All changes made in the revised version of our manuscript.

Yours.

Hani Alrefai

The corresponding author 

Reviewer 3 Report

Dear Editor of Biomolecules journal, 

I reviewed the manuscript biomolecules-1105418 entitled: “The antimicrobial, antioxidant, and anticancer activity of greenly synthesized selenium and zinc composite nanoparticles using Ephedra aphylla extract.” 

The paper is well written. Throughout the manuscript, some amended sections allowed a better understanding and solid support for the given information and the authors' results. After reading the work, I recommend the Editor publish it. Also, to improve the quality of the manuscript, I enlisted below some minor recommendations: 

In section 2.5.2, please write in cursive the species’ name Bacillus cereus.

Indicate in section 2.5.3 the reasoning behind selecting the cancer cell lines studied in their work.  Also, correct the number of cells seeded in the 96-well plate. Were the cell lines incubated for 48 h in a 96-well plate? If so, please explain whether, during this incubation time, a change in the cell population number could not affect the measurements. Normally in the cytotoxicity assays by MTT reduction, cells are incubated 24 h before exposed to NPs. Why do the authors decide to seed the cells for 48 h?

In figure 3, correct the legend to Comparison of the IC50 values of the tested samples against human cancer cells. 

To better understate the manuscript, I recommend transforming Table 3 and Table 4 into a cell viability graph, such as that presented in Figure 3.  

Author Response

Dear Reviewer, 3,

I am very glad to receive your letter, thanks for your time, effort and support.

Please find below our point-by-point response for the manuscript biomolecules-1105418. We have successfully replied to all of your precious comments and revised the manuscript accordingly. All changes made in the revised version of our manuscript.

Yours.

Hani Alrefai

This manuscript is a resubmission of an earlier submission. The following is a list of the peer review reports and author responses from that submission.

Round 1

Reviewer 1 Report

The research that you intend to publish is interesting, however, the manuscript needs further consideration prior to its acceptance. Therefore, I suggest that you recheck:

  1. the used terminology; there are several parts of the manuscript where the choice of words is not appropriate for the scientific use and make the text difficult to comprehend (e.g. see lines 48 - Green synthesis of nanomaterials by plant extracts contains...) Actually, by implies that the plant extracts is the method for the synthesis which is not true (the extract is part of the synthesis) and the green synthesis of nanomaterials doesn't contain anything, but the byproducts or the nanomaterials contain/include the active compounds from the plant. Other lines that need consideration are 68, 87, 92-93, 145-146, 246 etc
  2. what does "alternative phytochemical analysis" mean?
  3. lines 31-32 should contain the entire name of the microbial strains as this is their first appearance in the manuscript (this is to avoid any misunderstandings).
  4. you stated that "high potency as antioxidant agents as a result of 26
    the DPPH• assay." However, the results are not as promising as you state. There is a difference in intensity of 45 to 55 times less than the activity of the ascorbic acid (as per you declared values EC50 values of 27.77 and 34.65 mg extract / g DPPH that were almost comparable to ascorbic acid as a strong antioxidant with EC50 = 0.61 mg extract / g DPPH.)
  5. moreover, The obtained results indicated that Ephedra aphylla extract is a rich origin of phenolics flavonoids and tannins that possess reasonable antioxidant activity ...Using these groups that are responsible for the antioxidant activity of the extract in the biosynthesis leads to a decrease in the antioxidant activity.These affirmations are contradictory.
  6. The EC values represent a percent of inhibition and are expressed mg/gry weight or mg/mL of solution, refering to the extract and not to DPPH.
  7. The TEM microscopy confirms the existence of the nanoparticles but it also indicates the great variety of dimensions which will have a great impact on the repeatability and traceability of the tests. This is why without a stabilizer the obtained results are true exclusively for this batch of tests. 
  8. Did you take into consideration when you chose Zn as a chelator the fact that the complexes with flavonoids are difficult to break?
  9. You should try using Zn and Se as separate standard solutions, on their own on the cells, and take into consideration whether their activity is truly potentiated by the nanoparticles or not.  

Author Response

Dear Reviewer 1,

I am very glad to receive your letter, thanks for your time, effort and support.

Please find below our point-by-point response for the manuscript biomolecules-1058698. We have successfully replied to all of your precious comments and revised the manuscript accordingly. All changes made in the revised version of our manuscript.

Yours;

Hani Alrefai

The corresponding author

Reviewer 2 Report

The manuscript entitled “The antimicrobial, antioxidant, and anticancer activity of greenly synthesized selenium and zinc composite nanoparticles using Ephedra aphylla extract” presents an interesting issue. In this study, the findings addressed the composite nanoparticles of Ephedra aphylla extract combined with selenium and zinc exerted the anti-oxidant and cytotoxicity to tumour cells. The bioactive components extracted from ephedra aphylla mainly consist of ephedrine (a kind of alkaloid), the chemical structure was [r-(r*,s*)]-alpha-[1-(Methylamino) ethyl]-benzenemethanol, which have been used as anti-tussive, expectorant, anti-pyretic analgesic, and bronchodilator agents. Here, there are a critical issue that author should give a detailed information about the extraction procedure and molecular structure. However, in the present study it is confused that the author determining the phenolic, flavonoid, and tannin content, and no any information about the extract structure. Whether those compnents related to its bioactivities? if so, author should give an explanation about that. In the subsequent biological characterization, the author only listed the antioxidant, cytotoxic and antibacterial properties, but did not explore the relationship between these properties and the structure after selenization or zincization, which made the study very shallow, without scientific depth or mechanism exploration. I would not think it is publishable without new material supplementation or extensively editing.

Author Response

Dear Reviewer 2,

I am very glad to receive your letter, thanks for your time, effort and support.

Please find below our point-by-point response for the manuscript biomolecules-1058698. We have successfully replied to all of your precious comments and revised the manuscript accordingly. All changes made in the revised version of our manuscript.

Yours;

Hani Alrefai

The corresponding author

Reviewer 3 Report

I red the paper with great interest. The present work presents a green technique for the synthesis of selenium and zinc nanoparticles of the extract of Ephedra aphylla stems and the results of this study indicated that the prepared nano-solutions expressed potent antimicrobial and anticancer activities along with reduced antioxidant characters.

Author Response

Dear Reviewer 3,

I am very glad to receive your letter, thanks for your time, effort and support.

Yours;

Hani Alrefai

The corresponding author

Round 2

Reviewer 1 Report

I appreciate your answers point by point.

My concerns for point 9 is that we normale administer Zn and Se as antioxidants for usual therapy in their inorganic salt forms. Your research would have a greater impact if the green nanoformulation increases their (Zn, Se) health benefits by comparison to what it is already available for use. This is why I suggested using them as comparison standards. However, I know this usually signifies a long time to study and more expences. 

Reviewer 2 Report

I carefully read the revised and re-uploaded manuscript, but the author seemed unable to supplement some new data for the structural characteristics and bioactivities, so I don't think it is appropriate for publication in the current form.